# A Long-Term Observation on the Possible Adverse Effects in Japanese Adolescent Girls after Human Papillomavirus Vaccination

**DOI:** 10.3390/vaccines9080856

**Published:** 2021-08-04

**Authors:** Akiyo Hineno, Shu-Ichi Ikeda

**Affiliations:** 1Intractable Disease Care Center, Shinshu University Hospital, Matsumoto 390-0802, Japan; hineno@shinshu-u.ac.jp; 2Department of Medicine (Neurology and Rheumatology), School of Medicine, Shinshu University, Matsumoto 390-8621, Japan; 3Ikeda Medicine and Neurology Clinic, Azumino 399-8205, Japan

**Keywords:** human papillomavirus vaccination, adverse effects, orthostatic dysregulation, chronic regional pain syndrome, cognitive dysfunction

## Abstract

In Japan, a significant number of adolescent females noted unusual symptoms after receiving the human papillomavirus (HPV) vaccination, of which the vast majority of them were initially diagnosed with psychiatric illnesses because of the absence of pathologic radiological images and specific abnormalities in laboratory test results. Later these symptoms were thought to be adverse effects of HPV vaccination. However, a causal link between HPV vaccination and the development of these symptoms has not been demonstrated. Between June 2013 and March 2021, we examined 200 patients who noted various symptoms after HPV vaccination. In total, 87 were diagnosed with HPV vaccination-related symptoms based on our proposed diagnostic criteria. The clinical histories of these 87 patients were analyzed. The age at initial vaccination ranged from 11 to 19 years old (mean ± SD: 13.5 ± 1.5 years old), and the age at the first appearance of symptoms ranged from 12 to 20 years old (mean ± SD: 14.3 ± 1.6 years old). The patients received an initial HPV vaccine injection between May 2010 and May 2013, but the first affected patient developed symptoms in October 2010, and the last affected developed symptoms in October 2015. A cluster of patients with a post-HPV vaccination disorder has not appeared in Japan during the last five years. Our study shows that, in Japan, the period of HPV vaccination considerably overlapped with that of a unique post-HPV vaccination disorder development. This disorder appears as a combination of orthostatic intolerance, chronic regional pain syndrome, and cognitive dysfunction, but its exact pathogenesis remains unclear.

## 1. Introduction

The human papillomavirus (HPV) infection plays a crucial role in the development of uterine cervical cancers [1]. Therefore, in May 2010, HPV vaccines, Cervarix^®^ (GlaxoSmithKline, Brentford, UK), a papillomavirus recombinant bivalent vaccine, and Gardasil^®^ (Merck & Co, Inc., Kenilworth, NJ, USA), a papillomavirus recombinant quadrivalent vaccine, were widely introduced to Japanese female teenagers [2,3]. Beginning April 2013, female adolescents aged 13–16 years were legally required to receive this vaccination. Soon after this vaccination program began, a significant number of the vaccinated females complained of a unique disorder that was composed of violent tremulous involuntary movement, chronic pain, and weakness in the limbs. The Japanese mass media largely reported that a combination of these symptoms was previously unexperienced, suggesting that this disorder was a possible adverse reaction to HPV vaccination. Repeated presentations of suffering vaccinated females on television had a strong impact on Japanese society, forcing the Japanese Ministry of Public Health, Labour and Welfare to withdraw the recommendation for the use of HPV vaccination at the end of June 2013 [4]. Simultaneously, a special committee was organized to investigate the affected Japanese females, and our institution has been functioning as one of the investigation centers for the past eight years.

In our previous two reports [5,6], we described the clinical features and diagnostic criteria of the involved Japanese females with post HPV vaccination disorder. This disorder seems to include orthostatic dysregulation, chronic regional pain syndrome (CRPS), and cognitive dysfunction [5,6,7]. Post-vaccination abnormal autoimmune reactions are surmised to be responsible for this disorder [8,9], but a causal link has not been established between HPV vaccination and the appearance of these symptoms. Therefore, in this study, we attempted to clarify the temporal relationship between HPV vaccination and the development of this peculiar disorder on the basis of our single center’s long-term observation of the affected Japanese females.

## 2. Materials and Methods

Between June 2013 and March 2021, we examined the symptoms and objective findings of 200 HPV vaccinated female patients. According to our proposed diagnostic criteria [6], we obtained the necessary patient information, paying special attention to the duration between vaccination and the development of the first symptoms suspected to be related to the vaccine. The patients underwent physical and neurological examinations and routine laboratory tests. Skin temperature and a digital plethysmogram were recorded, and if necessary, the Schellong test was conducted. Moreover, neuropsychological tests and functional brain imaging were performed in patients with cognitive dysfunction. The details of these methods are described in our previous reports [5,6]. The study protocol was approved by the Institutional Review Board (approval nos. 4128 and 4150) of Shinshu University School of Medicine, Matsumoto, Japan.

## 3. Results

During the past eight years, 200 female patients visited our hospital with the suspicion of HPV vaccine-related adverse effects (33 patients in 2013, 43 in 2014, 38 in 2015, 49 in 2016, 25 in 2017, 8 in 2018, 4 in 2019, 0 in 2020, 0 in 2021). Of these, we excluded 19 patients who had symptoms before vaccination and 5 who received the HPV vaccine after 30 years of age. An additional 28 patients whose symptoms or disorders were explained by known diseases or who had abnormal laboratory data were also excluded, specifically, eight with epilepsy, six with psychiatric or anxiety disorders, three with systemic lupus erythematosus, one with juvenile idiopathic arthritis, one with anti-SGPS antibody-positive polymyositis, and nine with other diseases. For the remaining 148 patients, the clinical manifestations and objective findings were analyzed. The results showed that 32 patients were diagnosed with definite vaccine-related symptoms, and 55 were diagnosed with probable vaccine-related symptoms. The patient’s symptoms and signs of the 87 patients diagnosed are summarized in Table 1. The most frequent symptom was prolonged general fatigue, which led to an inability to wake up and subsequently go to school in the morning. Severe headache, widespread pain involving the limbs and trunk, and dysautonomic symptoms including orthostatic fainting and bowel dysfunction were also responsible for markedly decreased daily activity in the patients. Further, widespread pain typically appeared as migratory joint pain without any signs of inflammation, and intermittent neuralgic pain in the chest or abdominal wall was common. Motor dysfunction showed variable patterns, but the distal dominant weakness of the limbs, which was mimicking that of polyneuropathy, was predominant. Abnormal sensations were mainly observed in the thighs or lower legs where dysesthesia or allodynia was frequent. As compared with these symptoms, learning impairment and sleep disorder developed later. The patients complained of a lack of mental clarity. Objective findings that were frequently observed were orthostatic dysregulation, including postural orthostatic tachycardia syndrome (POTS), abnormal digital plethysmogram recordings, and abnormalities on brain SPECT images. The details of these findings have been described in a previous report [6].

The temporal distribution of the period of initial vaccination and the appearance of the first symptom in the diagnosed 87 patients is shown in Figure 1. Note that the initial vaccination period ranged from May 2010 to May 2013, and the age at initial vaccination ranged from 11 to 19 years old (mean ± SD: 13.5 ± 1.5 years old). Meanwhile, the first symptom appeared from October 2010 to October 2015, and the age at the appearance of the first symptoms ranged from 12 to 20 years old (mean ± SD: 14.3 ± 1.6 years old). Thus, the time from the first vaccine dose to symptom onset ranged from 0 to 1532 days (median: 199 days). The interval between the onset of symptoms and our initial examination ranged from 0 to 85 months (median: 31 months), indicating the illness duration in the patients before they visited our center.

The temporal relationship between the HPV vaccination and the development of the symptoms was as follows: the first HPV vaccine injection was in May 2010 and the last was in May 2013 (Figure 1a). The first affected vaccinated female developed symptoms in October 2010, and the latest appearance of symptoms occurred in two patients in October 2015; the peak period of the first injection of HPV vaccine seems to be between July 2011 and September 2012, and that of the development of unique post-vaccinated symptoms appeared between September 2011 and August 2013 (Figure 1a,b). Over the previous five years, we did not examine any patients who were newly affected by these unique symptoms (Figure 1b).

## 4. Discussion

HPV vaccine safety has been reported in HPV vaccination-predominant countries [10,11,12]. Especially in Australia, although syncope occasionally occurs after HPV vaccination, the frequency of other serious adverse effects including POTS, CRPS, primary ovarian insufficiency, Guillain-Barré syndrome, autoimmune diseases, and venous thrombosis is very low, suggesting no causal association [13]. However, the potential risk of HPV vaccination and dysautonomia, CRPS, and chronic fatigue syndrome has been identified based on a series of case reports from different countries [14,15,16,17,18,19]. Thus, safety concerns regarding HPV vaccines remain controversial [20].

According to the reports of a Japanese special committee [21,22], 3.39 million Japanese females received HPV vaccinations between May 2010 and November 2016, and 2024 recipients were reported to have adverse reactions, of which 673 experienced serious symptoms. However, the incidence of adverse reactions in this vaccination period was determined to be low and insignificant, even though similar symptoms were not observed as a result of other vaccines.

Variable clinical manifestations in the post-HPV vaccination disorders can be explained by a combination of orthostatic dysregulation, mainly appearing as POTS, CRPS, and/or cognitive dysfunction [5,6,23]. Recent research has found that among POTS, CRPS, and myalgic encephalomyelitis/chronic fatigue (ME/CFS), some conditions overlap [24,25,26]; especially for cognitive dysfunction, slow thinking, difficulty in focusing, lack of concentration, forgetfulness, and confusion are commonly observed in all three disorders, and correspond to haziness in thought process, which is currently called “brain fog” [27]. Thus, the cognitive dysfunction observed in patients with post-HPV vaccination disorders may be a secondarily induced pathological condition following the long-lasting POTS and/or CRPS. Furthermore, POTS, CRPS, and ME/CFS seem to share similar autoimmune abnormalities [28], and a few preliminary studies [29,30,31,32,33] and case reports [34,35,36,37] have shown that the presence of serum autoantibodies against autonomic nerve receptors may be a critical determinant in the pathogenesis of these three disorders. In relation to this hypothesis, we investigated the autoantibodies against autonomic nerve receptors in the serum of the affected patients and revealed that the serum levels of autoantibodies against the adrenergic receptors and muscarinic acetylcholine receptors were significantly elevated in patients with HPV vaccination, as compared with those in the controls [38]. However, there was no statistically significant association between the clinical symptoms and elevated serum levels of these autoantibodies. Thus, further studies are required to consider the possibility of HPV vaccination-related abnormal autoimmune reactions.

In our previous report [6], we described a close temporal relationship between HPV vaccine administration and the appearance of possible adverse symptoms in 72 Japanese patients on the basis of four years of observation. In this study, we extended this observation period to nearly eight years, and the number of patients diagnosed was increased to 87, reaffirming that the period of HPV vaccination considerably overlapped with that of a unique post-vaccination disorder development in our country. In Japan, HPV vaccine coverage for females aged 12 to 16 years has dropped to less than 1% after the termination of the government’s recommendation [39], and during the previous three years, few females visited us for evaluations regarding a suspected post-HPV-vaccination disorder. These observations indicate that intensive injections of HPV vaccines between May 2010 and May 2013 induced a cluster of Japanese patients with a unique post-HPV vaccination disorder. Japan is not the only exceptional country for an extremely lowered rate of HPV vaccination in recent several years; Latin American countries, such as Columbia, followed a similar pattern [40]. Adverse reactions to HPV vaccines seem to be influenced by different genetic backgrounds, cultural, and/or religious conditions. These conditions with no evidence of abnormal radiological images or laboratory data are often difficult to diagnose, easily leading to a pitfall of making the diagnosis of psychiatric illness.

Nevertheless, while there is a possible occurrence of adverse effects after HPV vaccination, these results do not necessarily signal the negation of the usefulness of this vaccine for the prevention of uterine cervical cancer [41]. If the information reported in this study is provided and is widely available at the induction of HPV vaccines, a social distaste for HPV vaccination (All Japan Coordinating Association of HPV Sufferers) would likely not occur in Japan. HPV vaccines are prophylactic and are not therapeutic, and thus, serious adverse effects are not acceptable, even if their incidences are low. Wide monitoring and an open discussion are recommended to ensure the safe announcement of HPV vaccines [42].

## Figures and Tables

**Figure 1 vaccines-09-00856-f001:**
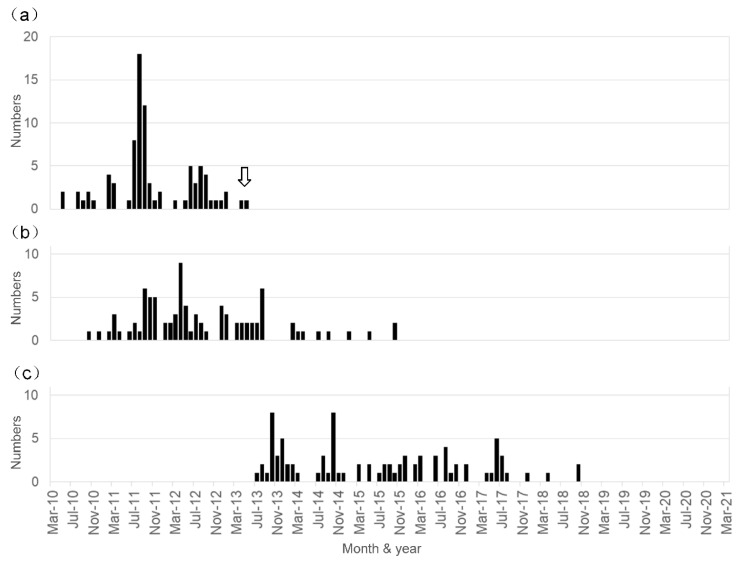
Temporal relationship between HPV vaccination and the development of symptoms in the patients diagnosed as having HPV vaccine-related symptoms. The period presented here ranged from May 2010 to March 2021. (**a**) Number of patients who received the first injection of HPV vaccine each month. The arrow indicates the time when the Japanese Ministry of Public Health, Labour and Welfare stopped the recommendation of HPV vaccination. (**b**) Number of patients who developed symptoms each month. (**c**) Number of patients who visited our institution and were diagnosed as having a post-HPV vaccination disorder each month.

**Table 1 vaccines-09-00856-t001:** Frequency of symptoms and signs in the 87 patients studied.

Symptoms	Number of Cases	Frequency (%)
General fatigue	73	83.9
Severe headache	72	82.8
Widespread pain	71	81.6
Dysautonomic symptoms	71	81.6
Motor dysfunction	56	64.4
Abnormal sensation	52	59.8
Learning impairment	52	59.8
Sleep disturbance	44	50.6
Menstrual abnormality	44	50.6
Limb shaking	41	47.1

## Data Availability

Not applicable.

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
