# Peer review of "A Long-Term Observation on the Possible Adverse Effects in Japanese Adolescent Girls after Human Papillomavirus Vaccination"

_vaccines, 2021, doi:10.3390/vaccines9080856_

Round 1

Reviewer 1 Report

There is little hesitancy in recommending that this excellent study, entitled "A long-term observation on the temporal relationship between human papillomavirus vaccination and its possible adverse effects in Japan " be published.  In so many respects, this study is remarkably important for future research in the field of HPV vaccines. The presentation on recent data regarding HPV vaccines despite the possible occurrence of adverse effects after HPV vaccination, does not necessarily signal the negation of the usefulness of this vaccine for the prevention of uterine cervical cancer.

Author Response

Dear Editors

Thank you very much for your reviewing our manuscript and giving excellent comments. According to your suggestions we have changed the manuscript in the following.

  1. Yearsyears old. More detailed information on patients’ examination were added in material and method.
  2. In discussion we clearly mentioned that intensive injections of HPV vaccines between May 2010 and May 2013 induced such a cluster of Japanese patients with a unique post-HPV-vaccination disorder on the basis of figure 1.
  3. In discussion the recent world situation of HPV vaccination is added: in Latin American countries including Columbia injection rate of this vaccine has been decreasing. Finally, it is emphasized that HPV vaccines are prophylactic and are not therapeutic and thus, serious adverse effects are not acceptable.

The manuscript has been re-checked by a specialist with native English-speaking colleague.

We hope that this revised manuscript is acceptable for publication in your journal.

With best regard.                            

Reviewer 2 Report

  • The age descriptions are confusing. When describing the absolute age, please use “years old” instead of years. On the other hand, it makes no sense to describe the age of the first appearance of symptoms ranged from 12-20 years (old) as the age of vaccination varies. It would be more appropriate to describe the symptomatic timeframe as post-vaccination time, e.g., on average, how long after vaccination did the patient developed vaccine-induced symptoms.
  • Figure 1 and the rationale behind the analysis are not well-illustrated, described, and discussed. The data was only “describe” in the result section without explaining the data, nor were the data fully discussed in the discussion section. The whole section needs much work before the message could be delivered to the reader.
  • In the current format, the discussion/conclusion is disconnected from the result section. The author should strive to make meaningful discussions based on the data reported instead of reporting self-narrative views/history logically disconnected from other parts of the manuscript.

Author Response

(The authors gave the same response as above.)
